# Hydrolysis of Extracellular ATP by Vascular Smooth Muscle Cells Transdifferentiated into Chondrocytes Generates P_i_ but Not PP_i_

**DOI:** 10.3390/ijms22062948

**Published:** 2021-03-14

**Authors:** Rene Buchet, Camille Tribes, Valentine Rouaix, Bastien Doumèche, Michele Fiore, Yuqing Wu, David Magne, Saida Mebarek

**Affiliations:** 1Institute for Molecular and Supramolecular Chemistry and Biochemistry, Université Lyon 1, French National Centre for Scientific Research, F-69622 Lyon, France; tribescamille@gmail.com (C.T.); valentine-rouaix.XD@orange.fr (V.R.); bastien.doumeche@univ-lyon1.fr (B.D.); michele.fiore@univ-lyon1.fr (M.F.); david.magne@univ-lyon1.fr (D.M.); saida.mebarek@univ-lyon1.fr (S.M.); 2State Key Laboratory of Supramolecular Structure and Materials, Institute of Theoretical Chemistry, Jilin University, Changchun 130012, China; yqwu@jlu.edu.cn

**Keywords:** ATP, alkaline phosphatase, aortic smooth muscle cell, calcification, chondrocyte, plaque calcification, kinetics, MOVAS, nucleotide, pyrophosphate

## Abstract

(1) Background: Tissue non-specific alkaline phosphatase (TNAP) is suspected to induce atherosclerosis plaque calcification. TNAP, during physiological mineralization, hydrolyzes the mineralization inhibitor inorganic pyrophosphate (PP_i_). Since atherosclerosis plaques are characterized by the presence of necrotic cells that probably release supraphysiological concentrations of ATP, we explored whether this extracellular adenosine triphosphate (ATP) is hydrolyzed into the mineralization inhibitor PP_i_ or the mineralization stimulator inorganic phosphate (P_i_), and whether TNAP is involved. (2) Methods: Murine aortic smooth muscle cell line (MOVAS cells) were transdifferentiated into chondrocyte-like cells in calcifying medium, containing ascorbic acid and β-glycerophosphate. ATP hydrolysis rates were determined in extracellular medium extracted from MOVAS cultures during their transdifferentiation, using ^31^P-NMR and IR spectroscopy. (3) Results: ATP and PP_i_ hydrolysis by MOVAS cells increased during transdifferentiation. ATP hydrolysis was sequential, yielding adenosine diphosphate (ADP), adenosine monophosphate (AMP), and adenosine without any detectable PP_i_. The addition of levamisole partially inhibited ATP hydrolysis, indicating that TNAP and other types of ectonucleoside triphoshatediphosphohydrolases contributed to ATP hydrolysis. (4) Conclusions: Our findings suggest that high ATP levels released by cells in proximity to vascular smooth muscle cells (VSMCs) in atherosclerosis plaques generate P_i_ and not PP_i_, which may exacerbate plaque calcification.

## 1. Introduction

Atherosclerosis calcification is a predictor of cardiovascular mortality [1]. Vulnerable plaques are characterized by a thin fibrous cap, macrophage infiltration, a large necrotic core, intraplaque hemorrhage, and calcification [2,3,4,5,6]. During atherosclerosis plaque calcification in mice, vascular smooth muscle cells (VSMCs) transdifferentiate into chondrocyte-like cells, which then induce calcification in the vascular walls [7,8,9]. Like osteoblasts and growth plate chondrocytes, these cells express bone/cartilage-specific Runt-related transcription factor 2 (RUNX2), triggering the expression of bone/cartilage-specific proteins, such as osteocalcin (encoded by *Bglap*) [10,11]. Moreover, these cells have increased tissue non-specific alkaline phosphatase (TNAP) activity and produce extracellular vesicles which accumulate calcium-phosphate crystals [11,12,13]. TNAP (E.C. 3.1.3.1) is a phosphomonoesterase with a broad range of substrates in vitro including nucleotides and pyrophosphate (Figure 1).

TNAP is the key enzyme during calcification since it hydrolyzes extracellular pyrophosphate (PP_i_), a strong inhibitor of apatite formation [14]. Extracellular PP_i_ originates from: (i) intracellular PP_i_ transported into the extracellular medium by progressive ankylosis protein homolog (ANKH) [15]; and (ii) from the hydrolysis of extracellular ATP by ectonucleotide pyrophosphatase phosphodiesterase (ENPP1) to produce PP_i_ and AMP (Figure 1) [16]. Under physiological conditions, normal levels of extracellular PP_i_ and other calcification inhibitors, including osteopontin, matrix-Gla proteins, and Fetuin A, are sufficient to prevent vascular calcification by inhibiting the formation of apatite. However, under pathological conditions, where the PP_i_ concentration can decrease, for example in the case of systemic ENPP1 deficiency (Generalized Arterial Calcification of Infancy), it is sufficient to induce vascular calcification [17]. Therefore, the P_i_/PP_i_ ratio regulates apatite formation [18,19]. Besides PP_i_, ATP and ADP can also inhibit apatite formation [20]. As a result of its ability to hydrolyze PP_i_, TNAP is a putative drug target to treat vascular calcification [21]. Under physiological conditions, the concentration of extracellular ATP is found to be in the 10–1000 nM range [22,23]. Upon atherosclerosis around a threefold (plus) increase in extracellular ATP is released from the cells [24]. Necrotic cells release high quantities of ATP that act as danger-signal recruiting leukocytes [25]. ATP is actively released from activated or stressed cells [26,27] and passively from necrotic cells [28] via ruptured cell membranes. Once released, ATP can bind to P2 purinergic receptors, triggering intracellular signaling [29]. Extracellular ATP is then rapidly hydrolyzed by TNAP [30], ENPP1 [16], and ectonucleoside triphoshatediphosphohydrolases, including CD39 (Figure 1) [31]. TNAP [30] and CD39 [31] dephosphorylate ADP to form AMP. AMP is further hydrolyzed by TNAP [13] and 5′-ectonucleotidase (CD73) [31]. VSMCs express several ecto-nucleotidases, including ENPP1, and can release ATP in a controlled manner [32,33,34]. Since high levels of ATP may lead to the generation of both P_i_ and PP_i_, we questioned whether this extracellular ATP is hydrolyzed into the mineralization inhibitor PP_i_ or the mineralization stimulator P_i_, and whether TNAP is involved. At present, very little is known about the hydrolysis of ATP by VSMCs, especially during transdifferentiation of VSMCs toward osteo-chondrocyte-like cells. Our aim was to measure the rates of ATP hydrolysis, to determine whether the ATP hydrolysis is sequential, and whether it can yield PP_i_ and to what extent. Furthermore, we estimated the number of ATP molecules hydrolyzed per cell and per minute to substantiate the time frame of the cellular response to extracellular ATP fluctuations. For this purpose, we selected murine MOVAS line cells as a cellular model of VSMC. An increase in the mRNA expression profile of key genes associated with vascular calcification, such as Akp2 (gene coding for TNAP), was observed in MOVAS-1 cells cultured under calcifying conditions, with similar changes in expression in murine aortic VSMCs [35]. MOVAS proliferate rapidly, have a long lifespan, and are reliable and convenient in vitro models of vascular calcification [35,36,37].

## 2. Results

### 2.1. Characterization of MOVAS Cells

To monitor the transdifferentiation of MOVAS cells toward the osteochondrogenic phenotype, we determined the presence of calcium phosphate and its TNAP activity. The quantity of calcium relative to the mass of proteins increased from day 0 to day 21 (Figure 2A), which correlates with a significant increase in specific TNAP activity from day 0 to day 21 (Figure 2B).

TNAP activity is a marker for calcifying cells [13,14]. Taken together, these findings confirmed that MOVAS cells were able to calcify during the transdifferentiation in osteogenic medium as reported earlier [35,36,37]. A culture of MOVAS cells in stimulation medium (ascorbic acid and β glycerophosphate) is a suitable model to investigate vascular calcification (VC) in vitro [35,36,37]. ^31^P-NMR and infrared spectroscopy were used to monitor the direct hydrolysis of ATP under physiological pH.

### 2.2. ^31^P-NMR Spectra of ATP Hydrolysis by MOVAS Cells during Transdifferentiation

Thanks to the natural abundance of phosphorous, ^31^P-NMR is a convenient method to quantify various nucleotides in a mixture (e.g., ATP hydrolysis products) without isolation (e.g., chromatographic analysis) or/and labeling [38]. Additionally, it does not require the use of radioactive species. Moreover, as a non-destructive method, it is particularly adapted to follow reaction kinetics [38]. Here, extracellular reaction media were extracted periodically from MOVAS to record ^31^P-NMR spectra from the whole reaction system, avoiding interference from intracellular nucleotides, nucleic acids, and phospholipids. MOVAS cells were incubated in osteogenic medium for 0, 7, 14, and 21 days. Then, after each incubation time in osteogenic medium, they were washed with Tris buffer (pH = 7.8), fixed with paraformaldehyde (PFA), and incubated in the presence of 50 mM extracellular ATP in Tris buffer with or without 5 mm Levamisole to determine the ATP hydrolysis rate. Extracellular media from fixed MOVAS cells were taken after 2, 24, 48, 72, and 96 h and analyzed by ^31^P-NMR. To illustrate the NMR findings, we selected those obtained on day 21 in the presence of 5 mM Levamisole (Figure 3).

ATP peaks (a triplet at −22.1 ppm, a doublet at −11.3 ppm, and a doublet at −6.6 ppm) decreased from 0 to 96 h, indicating significant hydrolysis of ATP (Figure 3). We observed the appearance of ADP (a doublet at −10.9 ppm and a doublet at −6.7 ppm) and P_i_ (a singlet at +2.2 ppm), which correlated with a decrease in ATP, indicating slight hydrolysis of ATP (Figure 3). ADP was hydrolyzed by forming AMP (a singlet at +3.5) and P_i_ (a singlet at +2.2 ppm). There was no formation of PP_i_ (the singlet at −7.45 ppm was not observed) ruling out the hydrolysis of ATP into AMP and PP_i_. ATP hydrolysis increased from days 0 to 21 (Appendix A). The presence of TNAP was evidenced by measuring its activity at alkaline pH (Figure 2B), with *p*-nitrophenylphosphate (*p*NPP) as substrate. The addition of 5 mM levamisole, an inhibitor of TNAP, decreased ATP hydrolysis (Appendix A). The hydrolysis of extracellular ATP, in the presence of levamisole, was not fully abolished, suggesting the occurrence of CD39 or various types of ectonucleoside triphoshatediphosphohydrolases in MOVAS cells. The determination of kinetic constants k_1_, k_2_, and k_3_ were obtained by assuming sequential hydrolysis of ATP, which was confirmed in the experimental data since no PP_i_ was observed (Figure 3). This was performed from each set of spectra measured at 0, 7, 14, and 21 days. The experimental data fitted well with the model (Appendix A) allowing the decrease in ATP to be quantified, together with the increase in ADP, followed by AMP and Pi (Table 1).

Regarding kinetic constants, k_1_ increased with the duration of incubation time from d9ay 0 to day 21, indicating a higher extracellular ATP hydrolysis rate, while k_2_ values, reflecting the extracellular ADP hydrolysis rate, were significant only from day 14 (Table 1). Under a saturating substrate concentration ([ATP] = 50 mM), the reaction rate was directly proportional to the enzyme concentration and therefore the kinetic rates reflect the phosphatase concentration associated with MOVAS. During maturation and differentiation, the number of MOVAS cells increased from approximately 0.97 million on day 0; 2.07 million on day 7; 3.5 million on day 14; to 5.1 million on day 21. The number of ATP molecules which were hydrolyzed per cell and per minute increased from 72 million on day 0; 143 million on day 7; 288 million on day 14; to 526 million on day 21 (Table 1, last column), corresponding approximately to a sevenfold increase in enzymes hydrolyzing ATP per cell, consistent with a 10-fold increase in specific TNAP activity (Figure 2B). The addition of 5 mM levamisole induced a decrease in the k_1_ and k_2_ values (Appendix A and Table 2) as compared to those obtained without levamisole (Appendix A, left panel and Table 1).

The first-order constant k_1_ was significant only after incubating MOVAS cells for 14 days, while k_2_ was significant only when MOVAS cells were incubated for 21 days. The activity on day 21 slowed down to 158 million molecules per minute and per cell, in the presence of 5 mM levamisole (Table 2, last column), indicating: (1) a partial inhibition of ATP hydrolysis by levamisole and (2) that about one-third of the phosphomonoesterase activity was due to ectonucleoside triphoshatediphosphohydrolases other than TNAP.

### 2.3. ^31^P-NMR Spectra of PP_i_ Hydrolysis by MOVAS Cells during Transdifferentiation

The hydrolysis of PP_i_ by MOVAS incubated in osteogenic medium for 0 and 15–17 days was determined by ^31^P-NMR, with the addition of 50 mM PP_i_ as substrate, under the same conditions as performed with 50 mM ATP. We observed significant PP_i_ hydrolysis induced by MOVAS, which increased from approximately 0.04 nmol min^−1^ on day 0 to 0.2 ± 0.1 nmol min^−1^ on day 15–17. One of the most likely enzymes for the PP_i_ hydrolysis is TNAP. Indeed, consistent with the fivefold increase in the hydrolysis of PP_i_ from day 0 to day 15–17, TNAP-specific activity in MOVAS increased 10-fold from day 0 to day 14 (Figure 2B, using *p*PNP as substrate at alkaline pH).

### 2.4. Infrared Spectra of ATP Hydrolysis by MOVAS Cells during Transdifferentiation

To complement the NMR findings, we analyzed the extracellular medium from PFA-fixed MOVAS using infrared spectroscopy. ATP (1231 cm^−1^), ADP (1212 cm^−1^), AMP (1088 cm^−1^), phosphate (1080 cm^−1^), and pyrophosphate (1107 cm^−1^) were identified from one of their respective phosphate vibration modes (Appendix A) [39,40]. For the sake of clarity, we present one typical example to illustrate the methodology. In this example, MOVAS cells were incubated in osteogenic medium for 14 days. Then, they were washed with Tris buffer, fixed, and incubated in the presence of 50 mm ATP in Tris buffer to determine the ATP hydrolysis by MOVAS cells. Extracellular media were collected after 3, 5, 24, and 48 h to be analyzed by IR. The IR spectrum of the medium after 3 h of incubation indicated the presence of ATP (Figure 4A, green trace).

ATP hydrolysis by MOVAS cells after 48 h (Figure 4A, black trace) was evidenced by a decrease in the 1231 cm^−1^ band corresponding to vibrational mode of phosphate belonging to ATP [39,40,41], with a concomitant increase in the band at 1080 cm^−1^ associated to the vibrational mode of phosphate [41]. IR difference spectra of extracellular media extracted from MOVAS cells incubated at the indicated times allowed us to determine the amount of phosphate as a function of time thanks to the molar extinction coefficient of P_i_ (Figure 4B). This calculation was done for each day of the transdifferentiation of MOVAS into chondrocyte-like cells. The rate was as low as 0.5 nmol min^−1^ on day 0 and increased to 6 nmol min^−1^ on day 21 (Table 3).

We observed an approximately twofold increase from 292 million phosphate produced per minute and per cell on day 0 to 625 million phosphate produced per minute and per cell on day 21 (Table 3, last column). We pooled activity values determined from NMR (Table 1) and from IR (Table 3) to evaluate the statistical significance of the increase (Figure 5). The increase in the ATP hydrolysis from day 0 was significant on days 14 and 21 (Figure 5A), while the increase in the activity per cell was significant on day 21 (Figure 5B).

To substantiate the sequential hydrolysis of ATP by MOVAS, we compared the IR difference spectra with the best-fitted curves composed of the IR difference spectra of the nucleotides (Figure 6).

The IR difference spectra corresponding to ATP hydrolysis (Figure 6, blue trace, ADP + P_i_ minus ATP), to ADP hydrolysis (Figure 6, violet trace, AMP + P_i_ minus ADP), and to AMP hydrolysis (Figure 6, orange trace, Pi minus AMP) served as input values to fit the IR difference spectra of the extracellular medium containing ATP measured after 48 h minus 3 h (Figure 6, full black line) and 24 h minus 3 h (Figure 6, full green line). The best fits (Figure 6, dashed lines) were not perfect, especially in the range 1060–960 cm^−1^, where neither nucleotide, P_i_ or PP_i_, absorbed, indicating contributions from other unidentified species. Nevertheless, the curve fitting was sufficient to confirm the sequential hydrolysis of ATP as observed on the dashed and full lines of the IR difference spectra for 48 h minus 3 h (Figure 6, black traces) and for 24 h minus 3 h (Figure 6, green traces). The experimental data were fitted with the model (Appendix A) allowing the decrease in ATP to be quantified, together with the increase in ADP, and AMP and Pi. We obtained a k_1_ value of approximately 0.01617 ± 0.00096 h^−1^ which was higher than that measured from the ^31^P-NMR. In addition to the sequential hydrolysis of ATP, as evidenced by the IR spectra, a shoulder at approximately 1107 cm^−1^ (Figure 4) may indicate the presence of PP_i_, which absorbs at 1107 cm^−1^ (Appendix A). AMP, which has a doublet at 1088 and 1107 cm^−1^ (Appendix A), may also contribute to the 1107 cm^−1^ shoulder. From the IR spectra, we cannot rule out the presence of PP_i_. However, the whole hydrolysis of ATP by MOVAS at a saturated ATP concentration is dominated by sequential phosphomonoesterase activity, which yields ADP, AMP, and P_i_, and not by the phosphodiesterase activity, which leads to PP_i_.

## 3. Discussion

### 3.1. Functions of Tissue Non-Specific Alkaline Phosphatase and Its Relevance to Vascular Calcification

We confirmed that the calcification of MOVAS-1 cells can be induced in the presence of calcifying medium (containing β-glycerophosphate and ascorbic acid) [35], as detected by Alizarin Red and the quantification of calcium deposition and alkaline phosphatase activity (Figure 2). In rodent MOVAS and A7R5 VSMCs, the addition of exogenous alkaline phosphatase (AP) or TNAP overexpression were sufficient to stimulate the expression of several chondrocyte markers and induce mineralization [42]. TNAP is an ectoenzyme that is necessary for mineralization due to its ability to hydrolyze PP_i_—a mineralization inhibitor—as observed in VSMCs transdifferentiated into chondrocyte-like cells, as well as in primary chondrocytes [42]. That was confirmed by the addition of levamisole, a TNAP inhibitor, which inhibited the PP_i_ hydrolysis in VSMCs transdifferentiated into chondrocyte-like cells [42]. Our findings indicated that the hydrolysis of exogeneous PP_i_ by MOVAS increased fivefold from non-mineralizing MOVAS to calcifying MOVAS after 15–17 days of incubation under calcifying medium. This is consistent with the postulate that activation of TNAP in VSMCs in vascular smooth muscle cells precedes vascular calcification [43].

### 3.2. Sequential Hydrolysis of ATP

Extracellular nucleotides are released from the secretory pathway or via conductive/transport mechanisms, such as membrane connexin hemichannels, pannexin channels, and ATP conducting anion channels (Figure 1), or when membranes lose their integrity as cells die through necrosis [32,44]. The hydrolysis of ATP by smooth muscle cells was sequential, indicating that the hydrolysis was dominated by phosphomonoesterase activity. It appears from the ^31^P-NMR data that more than two-thirds of the hydrolytic activity was inhibited in the presence of levamisole in fully trans-differentiated VSMC. This is consistent with previous reports that show that 75% of phosphatase activity in the plasma membrane is due to a levamisole-sensitive enzyme, probably TNAP [45]. Therefore, the remaining phosphomonesterase activity, including that of ectonucleoside triphosphate diphosphohydrolase 1 (CD39), which catalyzes the hydrolysis of ATP into ADP and ADP into AMP, cannot be excluded [31]. Furthermore, AMP can be further dephosphorylated to adenosine by 5′ ectonucleotidase or CD73 [31]. The hydrolysis of locally released ATP by ENPP1 is the major source of extracellular PP_i_ in chondrocytes and osteoblasts [46,47,48]. However, transdifferentiated smooth muscle cells did not produce detectable PP_i_ from ATP, suggesting that the phosphodiesterase activity was very weak, while PP_i_ is the major end product during hydrolysis of ATP by C6 rat glioma cells [49]. This indicates that non-mineralizing cells have a lower ability to hydrolyze PP_i_ as compared to calcifying cells, which is coherent with the observed fivefold increase in PP_i_ hydrolysis from non-calcifying MOVAS to calcifying MOVAS. This is also consistent with recent findings that report that pyrophosphate synthesis from ATP was reduced and pyrophosphate hydrolysis (via TNAP; PP_i_ →P_i_) was increased in both aortas and blood obtained from mice with Hutchinson–Gilford progeria syndrome [50]. Under pathological conditions, smooth muscle cells, by becoming calcifying cells, increased pyrophosphate hydrolysis. This indicates a possible treatment strategy for improving extracellular pyrophosphate metabolism, which could constitute an alternative therapy against vascular calcification with pyrophosphate deficiency [50].

### 3.3. Limitations and Comparisons between ^31^P-NMR and IR Findings

The transdifferentiation of smooth muscle cells into chondrocyte-like cells was accompanied by increasing specific TNAP activity determined under alkaline conditions with *p*NPP (Figure 2B), as well as increasing ATP hydrolysis by cells at neutral pH (Table 1 and Table 3). The increase in ATP hydrolysis by cells was not solely due to the increase in cell numbers (Figure 5) but also to the increase in phosphomonoesterase activity by a factor of two (IR findings, Table 3) to seven (^31^P-NMR findings, Table 1), confirming that transdifferentiation occurred. This was corroborated by the rises in calcium phosphate deposits and the amount of calcium during transdifferentiation (Figure 2A). ATP hydrolysis by MOVAS cells as determined from ^31^P-NMR was lower than that obtained from the IR spectra due to different experimental setups and variations. This is consistent with the fact that the decrease in ATP concentration was measured in the case of ^31^P-NMR, while the increase in P_i_ concentration, reflecting the hydrolysis of all nucleotides, was determined by IR. Moreover, the superposition of nucleotide signals in the IR leads to a more difficult data analysis while the signals in ^31^P-NMR are clearly defined. The fixation of cells by PFA may suggest but do not necessarily reflect all the action of cell-surface activities.

### 3.4. Physiological and Pathological Relevance of Kinetic Values Determined under Saturated Conditions

Under physiological conditions, extracellular ATP concentration can reach approximately 1–1000 nM [22,23,51,52], while pericellular ATP can reach approximately 1–10 μM [31]. Exogeneous ATP (50 μM) suppresses the calcification of living VSMC co-stimulated with cAMP and high P_i_ [33], suggesting that ATP (or other molecules) is sufficient to prevent calcification in living cells. Atherosclerotic plaques are characterized by the presence of necrotic macrophages and VMSCs, which are likely associated with dramatically increased local concentrations of ATP. In this context, our measurements were performed under the saturated concentration of 50 mM of ATP. The activity of the apparent ATP hydrolysis, which was approximately 72–292 pmol min^−1^ per million cells on day 0, was largely due to the high ATP concentration, as compared with those observed at physiological ATP concentrations, which were around a few pmol min^−1^ per million cells [53]. Such high concentrations of ATP would increase ATP hydrolysis to close to maximal values (Vm), affecting cellular activity. The addition of exogeneous ATP interacted with P2Y2 purinergic receptors, triggering TNAP expression, as evidenced in the osteoblasts [54], and may change CD39 and CD73 expression, as observed in rat cortical astrocytes [55]. The extracellular ATP hydrolysis reflected the overall contributions of all ecto-enzymes of the cells. The absence of any PP_i_ formation, even with low TNAP activity in smooth cells that are not fully transdifferentiated, is of particular interest. This was corroborated with the fivefold increase in the hydrolysis rate of PP_i_ from non-mineralizing MOVAS, which is reminiscent of normal physiological conditions, to calcifying MOVAS, which corresponds to pathological conditions, similar to those that occur in the media of arteries during aortic calcification. Indeed, previous reports indicate an association between the loss of plasma PP_i_ and vascular calcification [45,50,56,57]. PP_i_ deficiency contributes to vascular calcification [52]. Extracellular PP_i_ can be synthesized from extracellular ATP by ENPP1 or transported by ANKH, whereas PP_i_ is hydrolyzed to phosphate by TNAP, contributing to the formation of apatite crystals or calcium phosphate salts [56,57]. A small variation of PP_i_ concentrations within micromolar range has a drastic inhibition effect on apatite formation, in contrast to the millimolar range of P_i_ necessary to induce apatite formation [20]. Since the PP_i_ hydrolysis is dominated by TNAP this suggests that TNAP is a possible target to prevent calcification [56]. Taken together, these findings support that TNAP and other phosphatases in the vasculature contribute to the pathology of vascular calcification and that it is a druggable target [21,43,56,57,58,59].

## 4. Materials and Methods

### 4.1. Chemicals and Reagents

Culture media, serum, streptomycin, penicillin Alizarin red (ARS), bicinchoninic acid, *p*-nitrophenyl phosphate, Nonidet P-40, and cetylpyridimium chloride were obtained from Sigma Aldrich (Lyon, FR). Phosphate buffered saline (PBS) pH 7.4 contained 137 mM sodium chloride, 10 mM phosphate, and 2.7 mM potassium chloride (D8537, Sigma Aldrich^®^, Lyon, FR).

### 4.2. Cell Cultures

Mouse MOVAS cells from ATCC (ATCC^®^ CRL-2797, Molsheim, FR) were cultured in Dulbecco’s Modified Eagle’s Medium rich in glucose (4.5 g.mL^−1^) (DMEM, Sigma^®^, St. Louis, MO, USA) with L-glutamine (2 mM), streptomycin (0.1 mg.mL^−1^), penicillin (100 U), fetal bovine serum (10% *v*/*v*), and 4-(2-hydroxyethyl)-1-piperazineethanesulfonic acid buffer (20 µM). This medium is identified as the growth medium. Cultures were maintained in a humidified atmosphere consisting of 95% air and 5% CO2 at 37 °C. Cells were plated at a density of 10,000 cells.cm^−2^. Cells were grown confluent in growth medium supplemented with 50 μg.mL^−1^ ascorbic acid and 10 mM β-glycerophosphate (called osteogenic medium), for 0, 7, 14, and 21 days (respectively D0, D7, D14 and D21). The medium was changed three times per week. These two osteogenic factors enable MOVAS cells to become calcifying vascular cells [35,36,37]. Cell cultures were analyzed for calcium content (Section 4.3), protein determination (Section 4.4), and TNAP activity using visible spectrometry (Section 4.5), NMR (Section 4.6), and IR (Section 4.7).

### 4.3. Calcium Assay

At the indicated time of incubation (D0, D7, D14, and D21), the extracellular medium was removed and MOVAS cells were washed with 1 mL of PBS. Deposited calcium was extracted from the extracellular matrix using 200 µL of 0.6 M HCl and kept overnight at room temperature. Then, 20 µL of the supernatant was mixed with 100 µL of 2-amino-2-methyl-1-propanol and 100 µL of a mix of 160 µm *o*-cresolphtaleine complexone and 6.8 mm 8-hydroxyquinoline in water. It was incubated for 5 min at room temperature [36,37]. Absorbance was measured at 570 nm by TECAN^®^ Infinite M200 Pro microtiterplate reader. Calcium deposition was normalized to the protein amount in cells. Protein concentration was determined as indicated in Section 4.4.

### 4.4. Protein Determination

MOVAS cells grown at the indicated incubation times (D0, D7, D14, and D21) were harvested in 200 µL of 0.1M NaOH and 0.1% SDS aqueous solution for calcium determination or in 400 µL Nonidet P-40 (0.2%) for the determination of enzymatic activity. Cells were disrupted by sonication then centrifuged 5 min (2000 g) at 4 °C. Then, 10 µL of supernatant was completed to 200 µL of reactive medium as reported [36,37]. It was incubated for 30 min at 37 °C. The reaction was stopped by thermal shock (plunging into ice). Absorbance was measured at 562 nm by TECAN^®^ Infinite M200 Pro microtiterplate reader. Calibration was obtained using bovine serum albumin as standard.

### 4.5. Tissue Non-Specific Alkaline Phosphatase Activity Determination

MOVAS cells grown for the indicated incubation times (D0, D7, D14, and D21) were harvested in 0.2% Nonidet P-40 and disrupted by sonication then centrifuged 5 min (2000 g) at 4 °C. Cell lysates were used to determine TNAP activity to demonstrate the VSMC transdifferentiation associated with increased TNAP expression, which is a marker for mineral competent cells [32,33,35,36,42]. An aliquot (10 µL) of supernatant was pre-incubated for 5 min at 37 °C. Then, 190 µL of 10 mM *p*-NPP, 0.56 mM 2-amino-2-methyl-1-propanol and 1 mM MgCl_2_ in alkalinized distilled water (pH 10) was added [60]. The rate of *p*-nitrophenolate (*p*NP) release was measured for 2 min every 10 s at 405 nm by TECAN^®^ Infinite M200 Pro spectrophotometer. Specific activity was expressed as µmol of *p*NP released per minute and per milligram of protein (µmol.min^−1^.mg^−1^ or U.mg^−1^). Protein concentration was determined as indicated in Section 4.4.

### 4.6. Preparation of Extracellular Medium to Be Analyzed by ^31^P-NMR and IR

MOVAS cells were collected at the indicated time (D0, D7, D14, or D21) in growth medium with 50 μg.mL^−1^ ascorbic acid and 10 mM β-glycerophosphate, then they were washed with 1 mL of PBS, and fixed with 400 µL of 4% (*v*/*v*) PFA for 20 min at room temperature. PFA was then removed. PFA may affect the enzymatic activity of the cell surface. However, it was necessary to fix the cells with PFA since the incubation volume was too small to take aliquots of extracellular medium at time intervals. PFA or formalin were used to fix cells or tissues to determine activity or presence of ANKH [15], ENPP [17], and TNAP [17,42,58]. To initiate the phosphatase reaction, cells were incubated in Tris buffer (pH 7.8 100 mM Tris, 5 mM MgCl_2_, 5 µm ZnCl_2_) containing either 50 mM ATP or 50 mM PP_i_ corresponding to a total of around 0.5 mL including cells. The reaction was performed with or without 5 mM Levamisole, a commercial inhibitor of TNAP. At the indicated time (2, 5, 24, 48, 72, and 96 h), 50 µL of reaction medium was sampled and stored at −20 °C prior to NMR or IR analyses.

### 4.7. ^31^P-NMR Spectroscopy

Herein, 40 µL of reaction medium (see Section 4.6) was mixed with 40 µL of ^2^H_2_O (10% (*v*/*v*) in Tris buffer) and 320 µL of Tris buffer (pH 7.8, 100 mM Tris, 5 mm MgCl_2_, 5 µm ZnCl_2_,). Auto-shim was done on ^2^H_2_O. ^31^P-NMR spectra were measured with a Ascend™ 600 (Bruker^®^, Birrica, MA, USA) or with a 300 ultrashield (Bruker^®^) spectrometer. Simultaneous ATP, ADP, AMP, PP_i_, and P_i_ peaks were monitored [38]. Peaks were integrated to determine the concentrations of species based on the total phosphorous content (150 mM for), thanks to the natural abundancy of ^31^P. At least three independent measurements were performed to obtain kinetics parameters. In the absence of cells, the ATP hydrolysis was found to be negligible in the time course of the reaction.

### 4.8. Infrared Spectroscopy

IR spectra were acquired with a Thermo Scientific Nicolet iS10 spectrometer equipped with a DTGS detector. The IR spectra were recorded at 20 °C with 64 interferograms at 4 cm^−1^ resolution each and Fourier transformed. During data acquisition, the spectrometer was continuously purged with dry filtered air (Balston regenerating desiccant dryer, model 75-45 12 VDC). For infrared measurements, an aliquot of 5 µL of reaction medium (Section 4.6) was deposited between two CaF_2_ windows separated by a 12-μm spacers. It was further sealed in a temperature-controlled (25 °C) flow-through cell (model TFC M13-3 Harrick Scientific Corp.). The phosphate band located at 1080 cm^−1^ has a molar absorption coefficient of 1215 m^−1^ cm^−1^ [41]. At least three independent measurements were performed to calculate the phosphate release rate.

### 4.9. Kinetics Analysis

The kinetic model of ATP hydrolysis assumes a sequential and irreversible mechanism described by Equations (1)–(3):(1)$$TP+\;H_{2}O\overset{k_{1}}{\to }\;ADP+\;P_{i}$$
(2)$$ADP+\;H_{2}O\overset{k_{2}}{\to }\;AMP+\;P_{i}$$
(3)$$AMP+\;H_{2}O\overset{k_{3}}{\to }\;Adenosine+\;P_{i}$$
where *k*_1_, *k*_2_, and *k*_3_ are the corresponding apparent first-order kinetic constants. The theoretical model was curve-fitted using the DYNAFIT software [61], using time-dependent concentrations of nucleotides and phosphate, determined by ^31^P-NMR and IR. At least three independent measurements were performed to evaluate errors. Assuming that the K_M_ values of the phosphatases for ATP were below the mm range and because the substrate concentration was 50 mM, the rates could be assumed to be close to the V_m_ values. Therefore, the apparent first-order constants are strictly proportional to the phosphatase concentrations. To determine the ATP hydrolysis rate by the cells, V_m_ is expressed as nmol.min^−1^ using apparent first-order constant k_1_, neglecting variations of ADP and AMP from the ^31^P-NMR and IR spectra. Protein determination was performed according to Section 4.4.

### 4.10. Statistical Analysis

For each analysis in Figure 2, at least six independent experiments were performed. Groups were compared using Anova’s one-way test followed by Sidak’s multiple comparison test two-sided NMR (Table 1) and IR (Table 3) data (five independent experiments) were pooled and were subjected to Mann–Whitney test (Figure 5). Results were expressed as mean ± standard error of the mean (SEM). Results were considered significant when *p* < 0.05 (*), highly significant when *p* < 0.01 (**), and extremely significant when *p* < 0.001 (***).

## 5. Conclusions

The fact that MOVAS generated P_i_ and not PP_i_ is consistent with a possible molecular mechanism through which high ATP levels released by necrotic cells in atherosclerosis plaques generate P_i_, exacerbating plaque calcification.

## Figures and Tables

**Figure 1 ijms-22-02948-f001:**
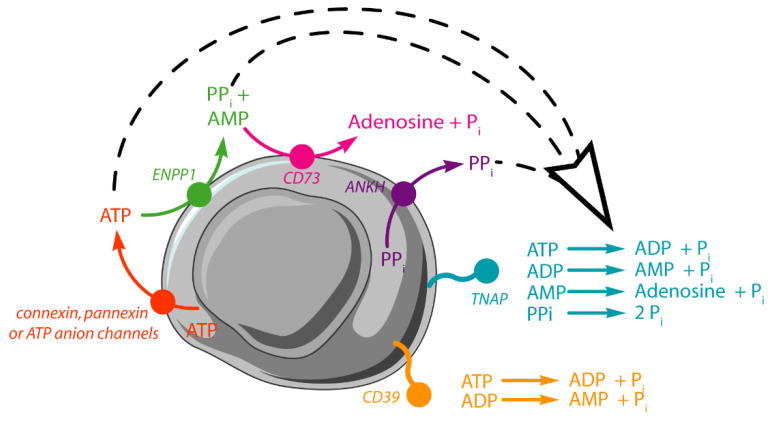
Extracellular adenosine triphosphate (ATP) is hydrolyzed by ectonucleotide pyrophosphatase phosphodiesterase (ENPP1) to form PP_i_ and adenosine monophosphate (AMP). AMP is further hydrolyzed by CD73 or tissue non-specific alkaline phosphatase (TNAP) to form adenosine and P_i_. ATP and adenosine diphosphate (ADP) can also be hydrolyzed by TNAP, CD39 (ENTPD1), or other types of ectonucleoside triphoshatediphosphohydrolases. The hydrolysis of PP_i_, which forms two P_i_, is mainly catalyzed by TNAP. ATP can be transported from intracellular to extracellular media via connexin, pannexin, or ATP anion channels, while PP_i_ is transported via progressive ankylosis protein homolog (ANKH). Eventually, large quantities of nucleotides are also released from necrotic cells.

**Figure 2 ijms-22-02948-f002:**
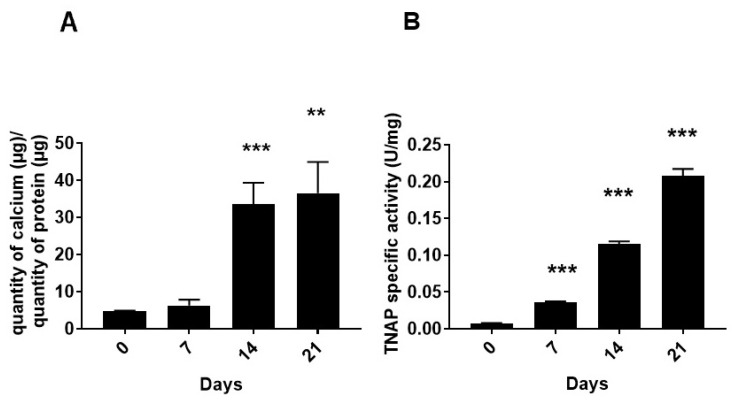
Characterization of the murine aortic smooth muscle cell line (MOVAS) during the transdifferentiation in osteogenic medium containing 50 µg.mL^−1^ of ascorbic acid and 10 mM of β-glycerophosphate. (**A**) Relative amount of calcium expressed as µg calcium per µg protein in calcium phosphate precipitates induced by MOVAS cells at the indicated incubation time in osteogenic medium (*n* = 6). (**B**) Tissue non-specific alkaline phosphatase activity in MOVAS cells at the indicated incubation time in osteogenic medium (*n* = 6). Results were considered highly significant when *p* < 0.01 (**), and extremely significant when *p* < 0.001 (***).

**Figure 3 ijms-22-02948-f003:**
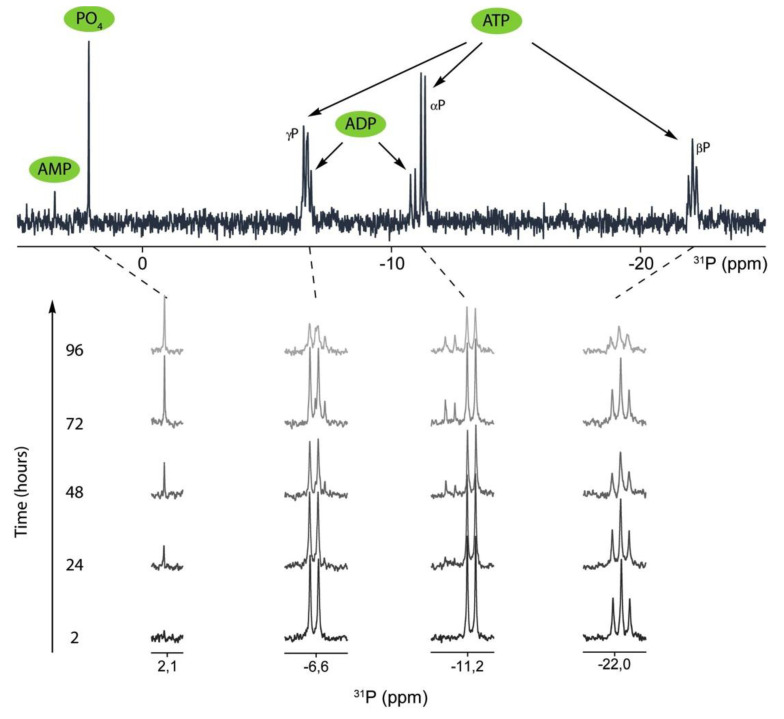
Hydrolysis of extracellular ATP by MOVAS cells incubated for 21 days in osteogenic medium. Thereafter, cells were washed, fixed, and incubated in Tris buffer at the indicated times of 0, 24, 48, 72, and 96 h in active medium containing 50 mM ATP. Then, 5 mM levamisole was added into the medium to inhibit TNAP. The absence of PP_i_ (no peak at −7.45 ppm) indicated that ATP hydrolysis by MOVAS cells was sequential and formed ADP and AMP.

**Figure 4 ijms-22-02948-f004:**
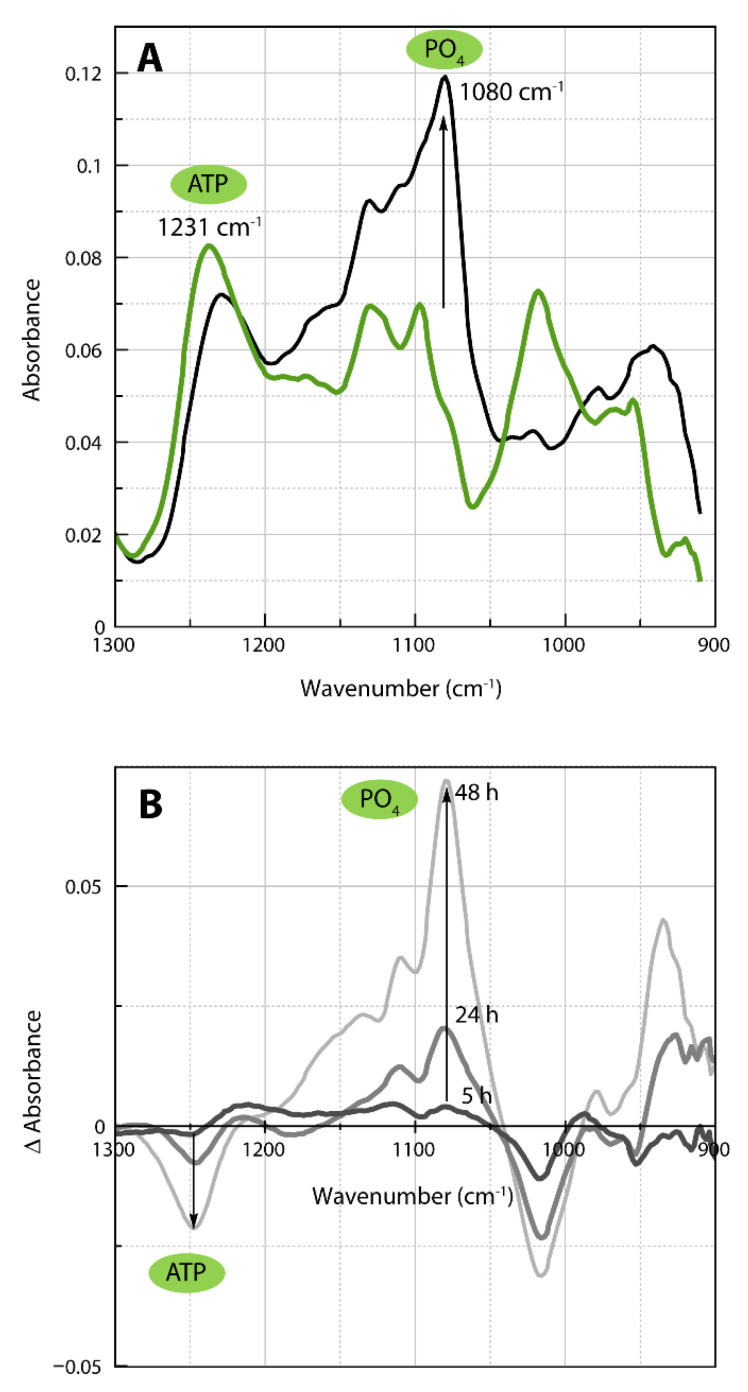
IR spectra of extracellular medium containing 50 mM ATP extracted from MOVAS after 14 days of incubation in osteogenic medium. (**A**) IR spectra obtained after 3 h (green) and 48 h (black) of ATP hydrolysis by extracellular phosphatases. (**B**) Successive IR difference spectra. ATP hydrolysis was monitored at the indicated incubation time. Dark gray: 5 h minus 3 h, medium gray: 24 h minus 3 h, light gray: 48 h minus 3 h.

**Figure 5 ijms-22-02948-f005:**
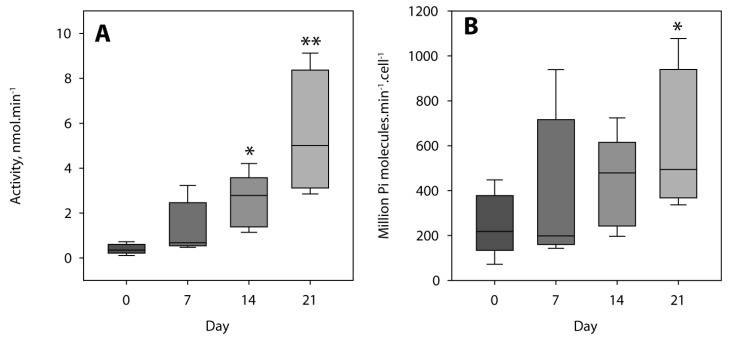
ATP hydrolysis by MOVAS cells expressed as (**A**) nmol min^−1^ and (**B**) as million P_i_ molecules min^−1^ cell^−1^ as determined by NMR (Table 1) and from IR (Table 3) (*n* = 5). Results were considered significant when *p* < 0.05 (*), and highly significant when *p* < 0.01 (**).

**Figure 6 ijms-22-02948-f006:**
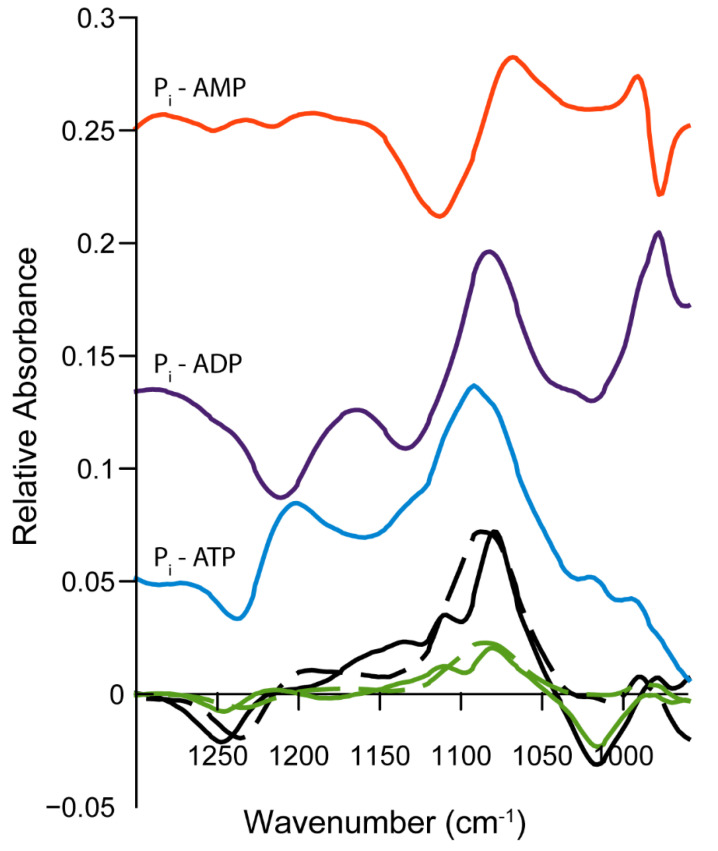
IR difference spectra of 50 mM of the nucleotides and of the extracellular medium containing 50 mM ATP extracted from MOVAS after 14 days of incubation in osteogenic medium. Orange: IR difference spectrum P_i_ minus AMP; violet: IR difference spectrum AMP + P_i_ minus ADP; blue: IR difference spectrum ADP + P_i_ minus ATP; Full black line: IR difference spectrum at 48 h minus 3 h; dashed black line: best fit; Full green line: IR difference spectrum at 24 h minus 3 h; dashed green line: best fit.

**Table 1 ijms-22-02948-t001:** Kinetic constants of extracellular ATP hydrolysis by MOVAS cells in the absence of levamisole on days 0, 7, 14, and 21 corresponding to the best fits (Appendix A).

Day	k_1_ (10^−3^ h^−1^)	k_2_ (10^−3^ h^−1^)	Activity (nmol ATP min^−1^)	Activity (Million ATP cell^−1^ min^−1^)
0	0.27 ± 0.07	0	0.11 ± 0.03	72 ± 19
7	1.14 ± 0.2	0	0.48 ± 0.08	143 ± 25
14	3.89 ± 0.2	1 ± 1	1.62 ± 0.08	288 ± 15
21	10.35 ± 0.4	6.1 ± 0.9	4.31 ± 0.17	526 ± 20

**Table 2 ijms-22-02948-t002:** Kinetic constants of extracellular ATP hydrolysis by MOVAS cells on days 0, 7, 14, and 21 corresponding to the best fits (Appendix A). After 0, 7, 14, and 21 days, 5 mM of levamisole was added just before the enzymatic assay.

Day	k_1_ (10^−3^ h^−1^)	k_2_ (10^−3^ h^−1^)	Activity (nmol ATP min^−1^)	Activity (Million ATP cell^−1^ min^−1^)
0	0	0	0	0
7	0	0	0	0
14	1.85 ± 0.2	0	0.77 ± 0.08	137 ± 15
21	3.11 ± 0.3	1.1 ± 1.6	1.30 ± 0.13	158 ± 15

**Table 3 ijms-22-02948-t003:** Kinetic constants of ATP hydrolysis by MOVAS cells on days 0, 7, 14, and 21, determined from the IR spectra. ATP hydrolysis was induced by all ectoenzymes of MOVAS cells, including TNAP.

Day	Activity (nmol min^−1^)	Activity (Million ATP per cell min^−1^)
0	0.5 ± 0.1	292 ± 57
7	1.6 ± 0.6	452 ± 177
14	2.8 ± 0.6	476 ± 108
21	6.0 ± 1.7	625 ± 198

## Data Availability

Not applicable.

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
