# Peer review of "Hydrolysis of Extracellular ATP by Vascular Smooth Muscle Cells Transdifferentiated into Chondrocytes Generates Pi but Not PPi"

_ijms, 2021, doi:10.3390/ijms22062948_

Round 1

Reviewer 1 Report

The authors have addressed all my concerns. My view is that the authors are aware of the limitations of the study in terms of correlation to actual physiology. In my opinion, the authors have addressed my concerns in a reasonable manner and I have no further queries.

Author Response

We thank the reviewer-1 for reviewing the manuscript and for positive comments. 

Reviewer 2 Report

The manuscript has been improved, but some minor correction is still needed:

  1. The One Way Anova Test followed by a post hoc test would be more appropriate than two-sided unpaired Student t-test to evaluate time dependent changes.
  2. Line 410-412: the authors state that Table 1, Table 2 and Table 3 were not subjected to two-sided unpaired Student t-test, was any other test used or has no statistical analysis been performed?

Reviewer 3 Report

The revised paper was an improvement. Hence, I think it is acceptable for publication.

Author Response

REVIEWER-3

The revised paper was an improvement. Hence, I think it is acceptable for publication.

Answer: We thank REVIEWER-3 for reviewing the manuscript and positive comments.

Reviewer 4 Report

Buchet et al. combined 31P-NMR and infrared (IR) spectroscopy techniques to investigate the hydrolysis of extracellular ATP in NOVAS cells trans-differentiated to chondrocyte-like cells. The working hypothesis is that, by hydrolyzing inorganic pyrophosphate (PPi, mineralization inhibitor), alkaline phosphatase (TNSP) favors the formation of the mineralization stimulator Pi, therefore, contributing to atherosclerosis plaque calcification. Major observations are: 1) Calcium phosphate accumulation, NASP activity (p-nitrophenol hydrolysis), and ATPase activity (against 50 mM exogenous ATP) increased during cell differentiation to chondrocytes. 2) NMR analysis indicated that ATP was metabolized to ADP and AMP, generating Pi as the final product. 3) ATP hydrolysis was partially inhibited by the TNSP inhibitor levamisol. 4) PPi accumulation was not detected during the incubation of cells with exogenous ATP. 5) Exogenous PPi was hydrolyzed by NOVAS cells. It is concluded that TNAP, a main PPi hydrolase, is a possible target to prevent vascular calcification. This is a meticulously performed study that uses state of the art spectroscopy approaches to assess nucleotide hydrolysis and metabolic product formation simultaneously.

I have the following major concerns.

  • An important conceptual concern is that there is no experimental evidence supporting that TNAP hydrolyzed PPi generated from the hydrolysis of ATP. Quite the opposite, Figures 3 and S1 show that no PPi formation occurred in the presence of levamisol (i.e., while TNAP was inhibited). These data strongly suggest that either:

a) An ecto pyrophosphatase activity (other than TNAP) hydrolyzed the PPi generated by eNPP. By the way, in the paragraph starting in L 171, we are told that exogenous PPi was significantly hydrolyzed by NOVAS cells. Was this hydrolysis inhibited by levamisol?

b) PPi was not generated at all, i.e., eNPPs are not functionally expressed in these cells.

In either case, the data suggest that TNAP plays no role in modulating the (extracellular) Pi:PPi ratio in these cells. The conclusions (Abstract) that TNAP is a possible therapeutic target to prevent vascular calcification should be revised.

  • Figure 2 assessed TNAP in a cell homogenate. However, ATP hydrolysis in intact cells is mediated by cell surface activities. To a what extent measurements of total cellular TNAP activity are relevant to cell surface TNAP? Why did the authors opt to use cell lysates rather than intact cells for the p-nitrophenol assay?
  • I am confused by the conditions (i.e., fixation) used to study ecto-ATPase activity. Why were the cells fixed (with PFA)? Are there precedents in the ATPase literature for this procedure? Importantly, how were these conditions validated to ensure that PFA did not affect the activity of cell surface ENTPDs, ENPPs, and TNAP? Why not using just rinsed intact/living cells?

Author Response

REVIEWER-4

Buchet et al. combined 31P-NMR and infrared (IR) spectroscopy techniques to investigate the hydrolysis of extracellular ATP in NOVAS cells trans-differentiated to chondrocyte-like cells. The working hypothesis is that, by hydrolyzing inorganic pyrophosphate (PPi, mineralization inhibitor), alkaline phosphatase (TNSP) favors the formation of the mineralization stimulator Pi, therefore, contributing to atherosclerosis plaque calcification. Major observations are: 1) Calcium phosphate accumulation, NASP activity (p-nitrophenol hydrolysis), and ATPase activity (against 50 mM exogenous ATP) increased during cell differentiation to chondrocytes. 2) NMR analysis indicated that ATP was metabolized to ADP and AMP, generating Pi as the final product. 3) ATP hydrolysis was partially inhibited by the TNSP inhibitor levamisol. 4) PPi accumulation was not detected during the incubation of cells with exogenous ATP. 5) Exogenous PPi was hydrolyzed by NOVAS cells. It is concluded that TNAP, a main PPi hydrolase, is a possible target to prevent vascular calcification. This is a meticulously performed study that uses state of the art spectroscopy approaches to assess nucleotide hydrolysis and metabolic product formation simultaneously.

I have the following major concerns.

1) An important conceptual concern is that there is no experimental evidence supporting that TNAP hydrolyzed PPi generated from the hydrolysis of ATP. Quite the opposite, Figures 3 and S1 show that no PPi formation occurred in the presence of levamisol (i.e., while TNAP was inhibited). These data strongly suggest that either:

  1. a) An ecto pyrophosphatase activity (other than TNAP) hydrolyzed the PPi generated by eNPP. By the way, in the paragraph starting in L 171, we are told that exogenous PPi was significantly hydrolyzed by NOVAS cells. Was this hydrolysis inhibited by levamisol?
  2. b) PPi was not generated at all, i.e., eNPPs are not functionally expressed in these cells.

Answer:

  1. We agree with REVIEWER-4 that there was no experimental evidence supporting that TNAP or other enzyme generated detectable PPi from the hydrolysis of ATP. Exogeneous PPi was hydrolyzed by MOVAS cells and partly inhibited by levamisole. This was stated in the discussion (marked blue) in:

Lines 281-283: However, trans-differentiated smooth muscle cells did not produce detectable PPi from ATP suggesting that the phosphodiesterase activity was very weak, while PPi is the major product end product during hydrolysis of ATP by C6 rat glioma cells [49].

Lines 261-263: Our findings indicated that the hydrolysis of exogeneous PPi by MOVAS increased by fivefold from non-mineralizing MOVAS to calcifying MOVAS after 15-17 days of incubation under calcifying medium.

Lines 272-276: It appears from the 31P-NMR data that more than two-third of the hydrolytic activity was inhibited in the presence of levamisole in fully trans-differentiated VSMC. This is consistent with previous reports showing that 75% of alkaline phosphatase activity in plasma membrane is due to a levamisole-sensitive enzyme, probably TNAP [35, 45].

  1. The amount of PPi was too small to be detected by NMR or by IR (although we can not exclude from IR that a small amount of PPi was present as stated in lines 247-250, blue marked ). eNPPs are expressed in smooth muscle cells [32-34] as stated in lines 83-84 blue marked :

Lines 247-250: From the IR spectra, we cannot rule out the presence of PPi. However, the whole hydrolysis of ATP by MOVAS at a saturated ATP concentration is dominated by the sequential phosphomonoesterase activity, yielding which yields ADP, AMP, and Pi, and not by the phosphodiesterase activity, leading which leads to PPi.

Lines 83-84: VSMCs express several ecto-nucleotidases, including ENPP1, and can release ATP in a controlled manner [32-34].

2) In either case, the data suggest that TNAP plays no role in modulating the (extracellular) Pi:PPi ratio in these cells. The conclusions (Abstract) that TNAP is a possible therapeutic target to prevent vascular calcification should be revised.

Answer: We agree with the REVIEWER-4 that other phosphatases may contribute to the hydrolysis of PPi.. However, TNAP has a role in the PPi hydrolysis as reported [56,58,59]. Indeed, small PPi concentration in the micromolar range has a drastic inhibition effect on apatite formation, while at least 1-2 mM of Pi is necessary to induce apatite formation in the presence of 1-2 mM Ca [20]. Since the PPi hydrolysis is dominated by TNAP this suggests that TNAP is a possible target to prevent calcification [56]. We modified the abstract and discussion to underline roles of other phosphatases.

Lines 31-34: Our findings suggest that high ATP levels released by cells in proximity to VSMCs in atherosclerosis plaques generate Pi and not PPi, which may exacerbate plaque calcification. TNAP and other phosphatases, as the main enzymes to hydrolyze PPi, is are a possible therapeutic targets to prevent vascular calcification.

Lines 333-338: Small variation of PPi concentrations within micromolar range has a drastic inhibition effect on apatite formation, in contrast to the millimolar range of Pi necessary to induce apatite formation [20]. Since the PPi hydrolysis is dominated by TNAP this suggests that TNAP is a possible target to prevent calcification [56]. Taken together, these findings support that TNAP and other phosphatases in the vasculature contributes to the pathology of vascular calcification and that it is a druggable target [58,59].

2)Figure 2 assessed TNAP in a cell homogenate. However, ATP hydrolysis in intact cells is mediated by cell surface activities. To a what extent measurements of total cellular TNAP activity are relevant to cell surface TNAP? Why did the authors opt to use cell lysates rather than intact cells for the p-nitrophenol assay?

Answer: The advantage of cell lysates is to obtain homogenous solution that make easy to determine TNAP activity. This is an established assay to monitor TNAP activity [17,32,33]. The activity has been determined at pH = 10, there are no other phosphatases than TNAP which can hydrolyze p-nitrophenolphosphate. Therefore the assay is highly specific of TNAP activity.

3) I am confused by the conditions (i.e., fixation) used to study ecto-ATPase activity. Why were the cells fixed (with PFA)? Are there precedents in the ATPase literature for this procedure? Importantly, how were these conditions validated to ensure that PFA did not affect the activity of cell surface ENTPDs, ENPPs, and TNAP? Why not using just rinsed intact/living cells?

Answer : We agree with the REVIEWER-4 that PFA may affect the activity of the cell surface. It was necessary to fix the cells since the incubation volume was too small to take aliquots of extracellular medium at time intervals. We added one paragraph in lines 388-390:

PFA may affect the enzymatic activity of the cell surface. However, it was necessary to fix the cells since the incubation volume was too small to take aliquots of extracellular medium at time intervals.

Round 2

Reviewer 4 Report

The Authors did not satisfactorily address my previous inquires and their responses contain statements that do not match with the text of the article. For example, in the authors responses, they refer to numbered lines in the revised manuscript that “are marked in blue”. I did not find any line in the entire manuscript that was marked in blue. I only found lines marked in yellow, which 1) do not correspond to the line numbers supposedly marked in blue, and 2) are not related to my questions. Furthermore, the text in most of the lines indicated by the Authors in their responses (L281-283, L261-263, L83-84, L247-250 etc.) do not correspond to those line numbers in the main manuscript.

The following major concerns remain unaddressed:

  1. There is not experimental evidence in this manuscript indicating that TNAP hydrolyzes PPi in these cells (i.e. no inhibition of PPi hydrolysis by levamisol was shown).   TNAP may play a role in plaque formation by directly generating Pi from ATP hydrolysis, but that PPi is also generated in these cells (and therefore could be target for plaque formation prevention) was not demonstrated.  
  2. My concern that the TNAP assay performed by the authors was in whole cell extracts and therefore does not reflect TNAP cell surface activity remains unanswered. The authors’ statement: “The advantage of cell lysates is to obtain homogenous solution that make easy to determine TNAP activityand is highly specific of TNAP activity” is not satisfactory. I do not dispute that this assay is specific for AP, but the relevance of these measurements to extracellular activities is questionable.
  3. My questions regarding PFA remain valid: How were the PFA (fixation) conditions validated to ensure that PFA did not affect the activity of cell surface ENTPDs, ENPPs, and TNAP? Why not using just rinsed intact/living cells? The authors’ statement: “PFA may affect the enzymatic activity of the cell surface. However, it was necessary to fix the cells since the incubation volume was too small to take aliquots of extracellular medium at time intervals” does not justify the approach. PFA may have profoundly altered the enzyme measurements.

Author Response

REVIEWER-4

Buchet et al. combined 31P-NMR and infrared (IR) spectroscopy techniques to investigate the hydrolysis of extracellular ATP in NOVAS cells trans-differentiated to chondrocyte-like cells. The working hypothesis is that, by hydrolyzing inorganic pyrophosphate (PPi, mineralization inhibitor), alkaline phosphatase (TNSP) favors the formation of the mineralization stimulator Pi, therefore, contributing to atherosclerosis plaque calcification. Major observations are: 1) Calcium phosphate accumulation, NASP activity (p-nitrophenol hydrolysis), and ATPase activity (against 50 mM exogenous ATP) increased during cell differentiation to chondrocytes. 2) NMR analysis indicated that ATP was metabolized to ADP and AMP, generating Pi as the final product. 3) ATP hydrolysis was partially inhibited by the TNSP inhibitor levamisol. 4) PPi accumulation was not detected during the incubation of cells with exogenous ATP. 5) Exogenous PPi was hydrolyzed by NOVAS cells. It is concluded that TNAP, a main PPi hydrolase, is a possible target to prevent vascular calcification. This is a meticulously performed study that uses state of the art spectroscopy approaches to assess nucleotide hydrolysis and metabolic product formation simultaneously.

I have the following major concerns.

1) An important conceptual concern is that there is no experimental evidence supporting that TNAP hydrolyzed PPi generated from the hydrolysis of ATP. Quite the opposite, Figures 3 and S1 show that no PPi formation occurred in the presence of levamisol (i.e., while TNAP was inhibited). These data strongly suggest that either:

  1. a) An ecto pyrophosphatase activity (other than TNAP) hydrolyzed the PPi generated by eNPP. By the way, in the paragraph starting in L 171, we are told that exogenous PPi was significantly hydrolyzed by NOVAS cells. Was this hydrolysis inhibited by levamisol?
  2. b) PPi was not generated at all, i.e., eNPPs are not functionally expressed in these cells.

Answer:

We thank  the reviewer-4 for its comments. The changes  were indicated by red, while blue marks indicated answers that did not require further changes in the manuscript.

  1. We agree with REVIEWER-4 that there was no experimental evidence supporting that TNAP or other enzyme generated detectable PPi from the hydrolysis of ATP. Exogeneous PPi was hydrolyzed by MOVAS cells and partly inhibited by levamisole. This was stated in the discussion (marked blue) in:

Lines 290-292: However, transdifferentiated smooth muscle cells did not produce detectable PPi from ATP suggesting that the phosphodiesterase activity was very weak, while PPi is the major product end product during hydrolysis of ATP by C6 rat glioma cells [49].

Lines 269-272: Our findings indicated that the hydrolysis of exogeneous PPi by MOVAS increased by fivefold from non-mineralizing MOVAS to calcifying MOVAS after 15-17 days of incubation under calcifying medium.

Lines 281-285: It appears from the 31P-NMR data that more than two-third of the hydrolytic activity was inhibited in the presence of levamisole in fully trans-differentiated VSMC. This is consistent with previous reports showing that shows that 75% of alkaline phosphatase activity in plasma membrane is due to a levamisole-sensitive enzyme, probably TNAP [35, 45].

  1. The amount of PPi was too small to be detected by NMR or by IR (although we can not exclude from IR that a small amount of PPi was present as stated in lines 247-250, blue marked ). eNPPs are expressed in smooth muscle cells [32-34] as stated in lines 83-84 blue marked :

Lines 255-258: From the IR spectra, we cannot rule out the presence of PPi. However, the whole hydrolysis of ATP by MOVAS at a saturated ATP concentration is dominated by the sequential phosphomonoesterase activity, yielding which yields ADP, AMP, and Pi, and not by the phosphodiesterase activity, leading which leads to PPi.

Lines 85-86: VSMCs express several ecto-nucleotidases, including ENPP1, and can release ATP in a controlled manner [32-34].

2) In either case, the data suggest that TNAP plays no role in modulating the (extracellular) Pi:PPi ratio in these cells. The conclusions (Abstract) that TNAP is a possible therapeutic target to prevent vascular calcification should be revised.

Answer: We agree with the REVIEWER-4 that other phosphatases may contribute to the hydrolysis of PPi.. However, TNAP has a role in the PPi hydrolysis as reported [56,58,59]. Indeed, small PPi concentration in the micromolar range has a drastic inhibition effect on apatite formation, while at least 1-2 mM of Pi is necessary to induce apatite formation in the presence of 1-2 mM Ca [20]. Since the PPi hydrolysis is dominated by TNAP this suggests that TNAP is a possible target to prevent calcification [56]. We modified the abstract and discussion to underline roles of other phosphatases.

Lines 31-34: Our findings suggest that high ATP levels released by cells in proximity to vascular smooth muscle cells (VSMCs) in atherosclerosis plaques generate Pi and not PPi, which may lead to exacerbate plaque calcification. TNAP and other phosphatases, as the main enzymes to hydrolyze PPi, is are a possible therapeutic targets to prevent vascular calcification.

Lines 343-349: Small variation of PPi concentrations within micromolar range has a drastic inhibition effect on apatite formation, in contrast to the millimolar range of Pi necessary to induce apatite formation [20]. Since the PPi hydrolysis is dominated by TNAP this suggests that TNAP is a possible target to prevent calcification [56]. Taken together, these findings support that TNAP and other phosphatases in the vasculature contributes to the pathology of vascular calcification and that it is a druggable target [58,59].

Round 3

Reviewer 4 Report

Specific comments:

The authors show that ATP hydrolysis is markedly -although not completely, inhibited by the TNAP inhibitor levamisol (compare Table 1 vs. Table 2). A corollary of this observation is that levamisol markedly reduces the release of Pi from adenine nucleotides. Importantly, the authors show that PPi was not formed in NOVAS cell, not even in the presence of levamisol (Fig. 3), suggesting that NPP/phosphodiesterases [i.e., ATP conversion to AMP + PPi] play a minor -if any, role in hydrolyzing ATP. Thus, no evidence is shown to support that PPi formation in NOVAS cells controls calcification. However, the authors state in the Discussion (L261-263) that “By degrading PPi, TNAP plays a pro-calcification role by altering the Pi/PPi ratio and promoting mineralization”. Furthermore, at the end of the Abstract, it is concluded that “TNAP and other phosphatases, as enzymes to hydrolyze PPi, are possible therapeutic targets to prevent vascular calcification”. These statements are confusing. Where is PPi coming from? While PPi may play a role in the calcification processes associated with Pi accumulation in VSMCs, the conclusion suggesting a role for PPi in the formation of Pi in NOVAS cells is premature and should be softened.

Regarding enzyme activity measurements, the assays described here may be suitable for histological studies, but whether the contribution of ecto- (cell surface) activities were rigorously and accurately assessed in cell homogenates and PFA-fixed preparations remains uncertain. Moreover, detection with antibodies (refs 15 and 17) do not prove that enzyme activities are preserved. The authors should acknowledge that the enzymatic activities measure in this study may suggest but do not necessarily reflect the action of cell-surface activities.

Author Response

The authors show that ATP hydrolysis is markedly -although not completely, inhibited by the TNAP inhibitor levamisol (compare Table 1 vs. Table 2). A corollary of this observation is that levamisol markedly reduces the release of Pi from adenine nucleotides. Importantly, the authors show that PPi was not formed in NOVAS cell, not even in the presence of levamisol (Fig. 3), suggesting that NPP/phosphodiesterases [i.e., ATP conversion to AMP + PPi] play a minor -if any, role in hydrolyzing ATP. Thus, no evidence is shown to support that PPi formation in NOVAS cells controls calcification. However, the authors state in the Discussion (L261-263) that “By degrading PPi, TNAP plays a pro-calcification role by altering the Pi/PPi ratio and promoting mineralization”. Furthermore, at the end of the Abstract, it is concluded that “TNAP and other phosphatases, as enzymes to hydrolyze PPi, are possible therapeutic targets to prevent vascular calcification”. These statements are confusing. Where is PPi coming from? While PPi may play a role in the calcification processes associated with Pi accumulation in VSMCs, the conclusion suggesting a role for PPi in the formation of Pi in NOVAS cells is premature and should be softened.

Answer: As suggested by the Reviewer, we softened the discussion, and the conclusion in the abstract. In the updated version, we deleted in the discussion L262-264 By degrading PPi, TNAP plays a pro-calcification role by altering the Pi/PPi ratio and promoting mineralization. It would thus make an attractive target for cardiovascular therapy [43].

We also deleted in the abstract L33-34: TNAP and other phosphatases, as enzymes to hydrolyze PPi, are possible therapeutic targets to prevent vascular calcification.

Regarding enzyme activity measurements, the assays described here may be suitable for histological studies, but whether the contribution of ecto- (cell surface) activities were rigorously and accurately assessed in cell homogenates and PFA-fixed preparations remains uncertain. Moreover, detection with antibodies (refs 15 and 17) do not prove that enzyme activities are preserved. The authors should acknowledge that the enzymatic activities measure in this study may suggest but do not necessarily reflect the action of cell-surface activities.

Answer: We acknowledged in the Materials and Methods in L387-389 that “PFA may affect the enzymatic activity of the cell surface. However, it was necessary to fix the cells with PFA since the incubation volume was too small to take aliquots of extracellular medium at time intervals.” (Text unchanged).

To make it clear, in the updated version we added: in the Discussion L303-304 “The fixation of cells by PFA may suggest but do not necessarily reflect the action of all cell-surface activities.”

Other additions were made at:

L123-126: Then, after each incubation time in osteogenic medium, they were washed with Tris buffer (pH = 7.8), fixed with paraformaldehyde (PFA), and incubated in the presence of 50 mM extracellular ATP in Tris buffer with or without 5 mM Levamisole to determine the ATP hydrolysis rate.

L185-186: To complement the NMR findings, we analyzed the extracellular medium from PFA-fixed MOVAS using infrared spectroscopy.

L 290: Limitations and comparisons between 31P-NMR and IR findings.

This manuscript is a resubmission of an earlier submission. The following is a list of the peer review reports and author responses from that submission.

Round 1

Reviewer 1 Report

This study addresses the process of extracellular ATP hydrolysis during trans-differentiation of vascular smooth muscle cells into chondrocytes. The underlying principle is that ATP can be hydrolyzed into ADP+Pi or AMP+PPi. PPi is a mineralization inhibitor, whereas Pi stimulates mineralization. This is important in the prognosis of atherosclerosis because mineralization triggered by calcification is a marker of poor prognosis and increased mortality.

The study itself is focused on the biochemistry of the model. First, they use the MOVAS cell line (these are immortalized mouse aorta/smooth muscle cells) to show that, using medium supplemented with ascorbate and beta-glycerophosphate, they produce calcium and TNAP, which are signs of calcification. Next, they address extracellular ATP hydrolysis by 31P NMR. They find all species but PPi, which leads them to the conclusion that ATP is hydrolyzed in a step-wise manner, first into ADP+Pi, and ADP into AMP+Pi. They conclude that most ATP hydrolysis is due to TNAP activity since incubation of the cells with the inhibitor levamisole abrogates most of the activity. They also analyze the IR spectrum of ATP hydrolysis in saturating ATP conditions (50mM), which confirms an increase in the Pi peak more or less consistent with the decrease in ATP.

This study is quite interesting and provide a probably correct answer to an important question in the field. However, there are unclear points and additional experimentation that has the potential to remove “probably” from “probably correct”. There are other issues with the manuscript that also need some clarification. These are described in the following point-by-point.

  • A typical problem with negative results is that additional proof is needed. If done by two techniques, a certain degree of uncertainty in one of them casts doubt over the whole hypothesis, which is the case here. The NMR data is clear enough, with no peak corresponding to PPi. However, IR spectra shows a very clear shoulder at the wavelength corresponding to pyrophosphate, around 1107 cm-1. This shoulder is not mentioned in the discussion of these data, but could indicate the presence of some PPi in the medium. This needs to be addressed.

  • The importance of these data is that there is an alternative hypothesis, that there is another extracellular pyrophosphate hydrolase that has preference for PPi as a substrate. One way of addressing this would be to add 32-P-labelled PPi to the cells and/or cell supernatants and study its disappearance. If the authors’ hypothesis is correct, the decline in the PPi levels should be comparable in conditioned vs. non-conditioned medium, indicating that there is no enzyme that cleaves PPi rapidly in these cells. If, on the other hand, PPi gets cleaved more rapidly, this means that MOVA cells secrete an ezyme that C6 rat glioma cells do not (ref. 47).

  • I find one column in Table 1 particularly disconcerting, which is the cell number. I assume the experiment is done side-by-side, and it is striking the numbers are downright identical, as if the inhibitor had no effect on the growth of the cells. This is likely a typo, but it needs to be corrected, because those numbers cannot be identical.

Reviewer 2 Report

In this study authors aimed to characterize the possible molecular mechanism by which hydrolysis of ATP occurs, especially during trans-differentiation of VSMCs toward osteo-chondrocyte-like cells. The products of ATP hydrolysis can be the mineralization inhibitor PPi or the mineralization stimulator inorganic phosphate (Pi), which have different role in atherosclerosis calcification, a predictor of cardiovascular mortality. The murine MOVAS line cells have been used as in vitro model of vascular calcification, demonstrating ATP hydrolysis is sequential yielding ADP, AMP and adenosine, generating Pi and not PPi. Moreover, results indicate the involvement of phosphatases including TNAP in ATP hydrolysis.

The study is interesting, but there is some concern about its description.

  1. High relevance is attributed to TNAP in introduction and results, while its role in the model is not enough described and clear in discussion.
  2. Overall a more in-depth discussion to clarify the meaning of the present results would be desirable.
  3. Some comments regarding translational implications should be added in the conclusion of both abstract and manuscript.
  4. The Student t-test has been used to evaluate statistical differences between groups, but it is not appropriate to evaluate time dependent changes.

Reviewer 3 Report

The authors showed that ATP hydrolysis by MOVAS cells was yielding the mineralization stimulator inorganic phosphate (Pi) but not the mineralization inhibitor pyrophosphate (PPi), and TNAP was involved in the hydrolysis ATP. MOVAS cells were cultured in calcification medium for 0-21 days. Then, the cells were fixed and incubated with a high concentration of ATP in the presence or absence of levamisole, followed by analysis of ATP metabolism by using NMR. The authors concluded that high ATP levels released by necrotic cells in proximity to VSMCs in atherosclerosis plaques generate Pi, which may lead to exacerbating plaque calcification. However, the experimental model is non-physiological and the conclusion is clearly an overstatement; 1) In vascular smooth muscle cells and aortas, not only TNAP (PPi -> Pi) but also eNPP (ATP -> PPi) and eNTPD (ATP -> Pi) are involved in the hydrolysis of ATP. The previous study (PNAS, 116 (47) 23698-704, 2019; doi.org/10.1073/pnas.1910972116) indicated that inhibition of TNAP or TNAP/eNPP increased production of PPi in ex vivo aortas. How is it consistent with the previous study? 2) It is well-known that necrotic cells release many DAMPs such as HSP and heparin sulfate in addition to ATP. Therefore, in the model of atherosclerosis plaques in the presence of necrotic cells proposed by the authors, the results of ATP alone are not sufficient.